# Relation of prior statin and anti-hypertensive use to severity of disease among patients hospitalized with COVID-19: Findings from the American Heart Association's COVID-19 Cardiovascular Disease Registry

Lori B. Daniels[1,2]*, Junting Ren[3‡], Kris Kumar[4], Quan M. Bui[1], Jing Zhang[5], Xinlian Zhang[3], Mariem A. Sawan[6], Howard Eisen[7], Christopher A. Longhurst[8], Karen Messer[3,5]

1 Division of Cardiovascular Medicine, Department of Medicine, UC San Diego, La Jolla, California, United States of America, 2 Division of Epidemiology, Department of Family Medicine and Public Health, UC San Diego, La Jolla, California, United States of America, 3 Division of Biostatistics and Bioinformatics, Department of Family Medicine and Public Health, UC San Diego, La Jolla, California, United States of America, 4 Knight Cardiovascular Institute, Oregon Health and Science University, Portland, Oregon, United States of America, 5 UCSD Moores Cancer Center, UC San Diego, La Jolla, California, United States of America, 6 Department of Medicine, Emory University School of Medicine, Atlanta, Georgia, 7 Pennsylvania State Health, Hershey, Pennsylvania, United States of America, 8 Department of Medicine, UC San Diego, La Jolla, California, United States of America

‡ Co-first author.
* lbdaniels@health.ucsd.edu

## Abstract

### Background

Statins have anti-inflammatory and immunomodulatory effects that may reduce the severity of coronavirus disease 2019 (COVID-19), in which organ dysfunction is mediated by severe inflammation. Large studies with diverse populations evaluating statin use and outcomes in COVID-19 are lacking.

### Methods and results

We used data from 10,541 patients hospitalized with COVID-19 through September 2020 at 104 US hospitals enrolled in the American Heart Association's COVID-19 Cardiovascular Disease (CVD) Registry to evaluate the associations between statin use and outcomes. Prior to admission, 42% of subjects (n = 4,449) used statins (7% on statins alone, 35% on statins plus anti-hypertensives). Death (or discharge to hospice) occurred in 2,212 subjects (21%). Outpatient use of statins, either alone or with anti-hypertensives, was associated with a reduced risk of death (adjusted odds ratio [aOR] 0.59, 95% CI 0.50–0.69), adjusting for demographic characteristics, insurance status, hospital site, and concurrent medications by logistic regression. In propensity-matched analyses, use of statins and/or anti-hypertensives was associated with a reduced risk of death among those with a history of CVD and/or

**Data Availability Statement:** The data are third party data. We did not receive any special access

privileges that others would not have; indeed, others will be able to access the data in the same manner as us. A non-author contact from the American Heart Association that interested researchers can get in touch with in terms of accessing the data is Heather Alger, who can be reached at Heather.alger@heart.org.

**Funding:** The author(s) received no specific funding for this work.

**Competing interests:** The authors have declared that no competing interests exist.

hypertension (aOR 0.68, 95% CI 0.58–0.81). An observed 16% reduction in odds of death among those without CVD and/or hypertension was not statistically significant.

## Conclusions

Patients taking statins prior to hospitalization for COVID-19 had substantially lower odds of death, primarily among individuals with a history of CVD and/or hypertension. These observations support the continuation and aggressive initiation of statin and anti-hypertensive therapies among patients at risk for COVID-19, if these treatments are indicated based upon underlying medical conditions.

## Introduction

The severe acute respiratory syndrome coronavirus 2 (SARS-CoV-2) is responsible for the COVID-19 pandemic which is causing worldwide morbidity and mortality. Incident cases, hospitalizations, and deaths from COVID-19 remain widespread, despite improvements in research methods and treatment regimens, and the recent rollout of vaccines. Understanding factors that mitigate the severity of COVID-19 remains critically important. Previous observational studies have found that statins may reduce the severity of COVID-19 infection [1–4]; however these studies have been limited in size with mostly single-center or regional studies, and some results have been conflicting [5]. Furthermore, statins are used both by sicker individuals with underlying cardiovascular disease (CVD) and hypertension, and by healthier individuals for prevention of CVD. Whether any protective effect of statins on COVID-19 outcomes is modified by underlying health conditions is unknown, in large part due to sample size limitations of previous studies. In particular, it is of great interest to understand if statins are equally protective in healthy individuals, as this information would have both clinical and mechanistic implications. Several mechanisms have been proposed for a potential beneficial effect of statins in the setting of COVID-19, including a potential cardioprotective effect that limits myocardial injury which some patients develop; via a direct inhibitory effect on the virus [6]; and/or via pleiotropic properties of statins including anti-inflammatory effects, immune system modulation, reduction of oxidative stress, and improvement in endothelial cell function [7]. The American Heart Association (AHA) COVID-19 CVD Registry systematically collected hospitalized patient-level data in a broad and diverse hospital and patient population across the United States (U.S.) [8]. Using these data, we sought to comprehensively evaluate the association of prior outpatient statin therapy on the severity of infection among a large cohort of patients hospitalized for COVID-19.

## Materials and methods

### Patient population

The AHA COVID-19 CVD Registry has been previously described [8]. Briefly, the Registry, powered by Get With The Guidelines®, includes all consecutive adult patients hospitalized from January 2020 through September 2020 with active COVID-19 disease confirmed by RT-PCR test or positive IgM antibody test, at any of 104 participating hospitals regardless of CVD status. Patients without active infection (determined by COVID-19 related symptoms) were excluded. Patient data were extracted from the medical record retrospectively, including dates of admission and discharge, age, sex, and medical history; the data were not included in

the registry until the patient was discharged from the hospital. Participating hospitals are provided detailed data abstraction instructions, and in-form and post-collection data quality checks are performed. The registry includes 15,673 admission records (15,397 unique patients). For the present study, we sequentially excluded patients who were below age 40 years (2,062 records), those whose COVID-diagnosis date was after their discharge date (319 records), those who were transferred from another hospital (2,671 records), and those with unknown final disposition (80 records; Fig 1). The remaining 10,541 records (10,335 unique patients) comprised the patient population for the primary outcome (incidence of severe disease). For the competing risk analyses (time to severe disease or recovery), records with missing date of COVID-19 symptom onset (1,574 records) or date of development of severe disease (39 records) were excluded, leaving 8,928 records (8,772 unique patients). Hereafter, we will refer to records as subjects or patients. Because the AHA COVID-19 CVD Registry is designed as a quality improvement tool, with no intervention or participant contact, and collection of only a limited data set, informed consent is not obtained. IQVIA (Parsippany, NJ) serves as the data collection center and the Duke Clinical Research Institute is the data-coordinating center for the registry. The Duke University Institutional Review Board provided approval.

## Study measures

The AHA COVID-19 CVD Registry includes over 200 data elements, including patient demographics, insurance status, medical history including outpatient medications, presenting symptoms including their date of onset, admission and discharge dates, and outcomes. The case report form is available online [9]. There is no code for missing data in the registry, so subjects with missing data are generally assumed to be negative for presence of the measured item.

Our *a priori* exposure of interest was documented outpatient use of statin medication at the time of hospital admission. However, 83% of those on statins were also on anti-hypertension medication (anti-HTN) including ACEi, ARB, calcium channel blockers, diuretics, and beta blockers, which may also influence outcomes in COVID-19 patients. Therefore, we investigated an indicator of exposure to these two groups of medications, with the mutually exclusive categories use of anti-HTN alone, use of statin alone, use of both statin and anti-HTN, medications, and use of neither category.

Race/ethnicity was by self-report, and was divided into Non-Hispanic White, Hispanic, Black, and Other. Insurance status was divided into unknown/none, Medicare, private, and public insurance. Obesity was defined as body mass index over 30 kg/m$^2$. Hypertension was defined as physician diagnosed high blood pressure, regardless of treatment. Diabetes was defined as a history of confirmed physician diagnosed diabetes mellitus (type I or II), or treatment for diabetes including the use of diet, oral hypoglycemic agents or insulin. Dyslipidemia was defined as a history of high cholesterol, hyperlipidemia or hypercholesterolemia based on physician diagnosis, treatment with a lipid lowering agent, total cholesterol greater than 200, LDL greater than 100, HDL less than 40, or elevated triglycerides greater than 200. Chronic kidney disease (CKD) was defined as a history of physician diagnosed renal insufficiency or chronic failure or if the serum creatinine was greater than 2.0mg/dl. CVD was defined as history of myocardial infarction or coronary revascularization (percutaneous intervention or bypass surgery), transient ischemic attack or stroke, peripheral arterial disease, heart failure, or atrial fibrillation or flutter. Comorbidities included a history of obesity, hypertension, diabetes, dyslipidemia, CKD, CVD, cancer, immune disorders, smoking or vaping, and pulmonary disease. For comorbidities with less than 5% prevalence we made a pooled indicator variable for the presence of one or more of these conditions.

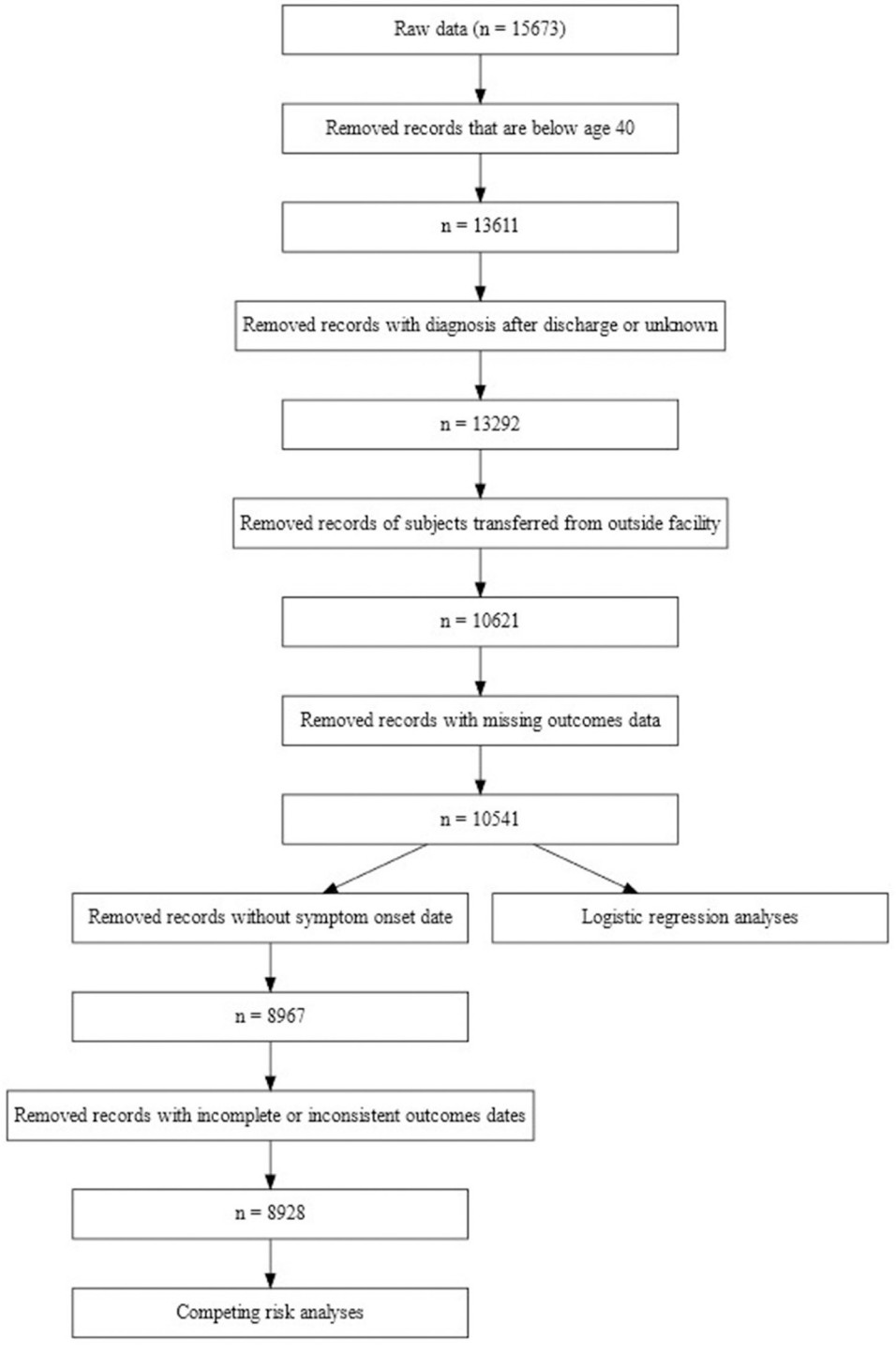

**Fig 1. Flow chart of study population.** Flow chart showing disposition of patient population.

## Outcomes

The primary outcome was in-hospital all-cause death or discharge to hospice care (hereafter referred to as "death"). The secondary outcome was the development of severe disease,

defined as admission to the intensive care unit (ICU), need for intubation and/or mechanical ventilation, or in-hospital death or discharge to hospice care. Patients discharged from the hospital without experiencing a severe outcome were defined as recovered from COVID-19.

## Statistical analysis

For each predictor of interest, means (standard deviation [SD]) or proportions are presented, stratified by presence or absence of severe disease. The odds ratio for the use of statin and/or anti-HTN based upon patient characteristics and comorbidities was computed, along with a 95% confidence interval (CI), using Fisher's exact test. As recommended, analyses are conducted per patient record [10].

Because presence or absence of CVD and hypertension is likely to be strongly associated with both the exposure of interest and the development of a severe outcome with COVID-19, and because presence of CVD and hypertension may be effect modifiers of statins, we investigated the effect of medication exposure separately within patient groups stratified by health status using a propensity score-matched analysis: among all patients with a history of CVD and/or hypertension, we first used logistic regression to model the propensity for use of any of the medications of interest (statin and/or anti-HTN), and then matched each unexposed subject with up to two exposed subjects, without replacement, on the resulting propensity score using a caliper width of 0.10 (R package *matchit*). Matching variables included age, sex, race, insurance status, admission month, hospital site, use of antiplatelet medications, use of anticoagulant medications, as well as indicators for the comorbidity measures listed above, all of which were considered *a priori* to be potentially related to both severe disease and use of medications of interest. The proportion of subjects who died was then compared between exposed and unexposed patients for the matched sample. This procedure estimates the causal effect of medication use (statin or anti-HTN medication), among the group of subjects with CVD and/or hypertension who do not use medication and who can be well-matched to similar subjects who do use medication. This approach is less dependent on modeling assumptions than an analysis which uses covariate adjusted regression-based estimates for the combined population. Finally, among patients with no documented history of either CVD or hypertension, a similar procedure was used, except we matched each exposed subject to two or more unexposed subjects, thus estimating the causal effect of statin or anti-HTN medication use within this relatively healthy group who were on medication and could be well-matched to a subject not on medication. For the secondary outcome of severe COVID-19, the same methods were applied. Confidence intervals and p-values were constructed conditional on the matched samples.

Additional analyses using multivariable mixed effects logistic regression (*glmer* from the R package *lem4)* were performed in order to investigate the overall mean effect of statins in the study population at large, without stratification by underlying conditions. In these models the outcome was death (or severe disease), the predictor of interest was use of statins and/or anti-HTN medication modeled as a four level variable, and covariates included age, sex, race, insurance status, use of antiplatelet medications, use of anticoagulant medications and presence of each of the comorbid conditions listed above. A fixed effect for admission date was modeled using a natural cubic b-spline by admission month, with two knots chosen at tertiles. Hospital sites were included as a random intercept.

As a sensitivity analysis a competing-risks analysis was used to investigate the association of the exposure of interest (use of statin or anti-HTN) with time to onset of the first of either severe disease or recovery. Details are provided in the S1 Appendix. All analyses were

conducted using R v3.6.0 statistical software. Two-tailed p-values <0.05 were considered statistically significant.

## Results

### Patient characteristics

The primary study population included 10,541 subjects hospitalized with COVID-19 disease, including 2,212 (21%) who died and 4,161 (39%) who experienced a severe outcome. ICU admission incidence overall was 30%, and 19% required mechanical ventilation.

Overall, 55% of subjects were male; the mean age was $66 \pm 14$ years. About one third of the subjects (36%) were non-Hispanic white, with the remainder Hispanic (26%), Black (26%), or other and/or mixed race (12%). Just over half (53%) were on Medicare, 20% were on public insurance, 21% on private insurance, and 7% had either no insurance or were indeterminant. Among comorbid conditions, 66% had a history of hypertension, 41% of patients were obese, 40% had diabetes, 40% had dyslipidemia, and 32% had CVD. Other comorbid conditions with 5% or greater prevalence included pulmonary disease (18%), CKD (15%), cancer (12%) and history of smoking/vaping (7%). Among the 3,360 CVD patients, 83% also had a history of hypertension.

### Use of statins and anti-hypertensive medications

Most patients (71%) had either CVD, hypertension, or both. Of these, 85% were taking a statin and/or anti-HTN. Among patients with neither CVD nor hypertension, only 19% were taking one of more of these medications. In the overall population, 82% of subjects on statin medication were also taking at least one anti-HTN: 35% of subjects overall used both statin and anti-HTN, 7% used statins alone, 24% used anti-HTN alone and 34% used neither.

Compared to those not taking either class of medication, patients taking statins and/or anti-HTN were older, more likely to be female, more likely to be on anti-platelet agents and anticoagulants, and more likely to have a number of comorbidities including CVD, hypertension, diabetes, cancer, CKD, pulmonary disease, and obesity (Table 1). Hispanics and those of "Other" race/ethnicity were less likely to be taking statins and/or anti-HTN compared to non-Hispanic Whites.

### Univariable associations with death and severe COVID-19

Patients who died were more likely to have CVD and to have hypertension than those who survived (S1 Table). They were also more likely to be male, older, non-Hispanic White, with public insurance, and were more likely to have a history of diabetes, cancer, CKD, dyslipidemia, and pulmonary disease. They also were significantly more likely to be on statins (odds ratio 1.24, 95% CI 1.13 to 1.37). Use of statins alone and anti-hypertensives alone were each a marker for increased risk of death, which occurred in 18% and 22%, respectively. Among patients on both classes of medications, 24% died, compared with only 17% of those on neither class of medication (Pearson's chi square test, p<0.01; Table 2).

When the secondary outcome of severe COVID-19 was considered, a similar pattern was seen: severe outcome occurred in 46% of those with CVD alone, 40% of those with hypertension alone, and 47% with both conditions, compared to 30% of those with neither condition (Pearson's chi square test, p<0.01). Patients with severe COVID-19 were more likely to be on statins (odds ratio 1.15, 95% CI 1.06 to 1.24, S2 Table). Among those on statins alone, anti-hypertensives alone, or both, 38%, 42%, and 42% had a severe outcome, respectively, compared to 35% of those on neither class of medication (Pearson's chi square test, p<0.01).

**Table 1. Characteristics of patients on statin and/or anti-hypertensive medications compared to those on neither medication.**

|  | No statin/anti-HTN (n = 3598) | On statin and/or anti-HTN (n = 6943) | OR | 95% CI | p |
|---|---|---|---|---|---|
| **Male** | 2,125 (59%) | 3,690 (53%) | 0.79 | 0.72,0.85 | <0.001 |
| **Age (years)\*** | 60.3 ± 14.0 | 69.3 ± 13.2 | 1.64 | 1.58,1.69 | <0.001 |
| **Insurance status** |  |  |  |  |  |
| Medicare (over age 65) | 1,207 (34%) | 4,355 (63%) | 3.53 | 3.17,3.93 | <0.001 |
| Public | 833 (23%) | 1,231 (18%) | 1.45 | 1.28,1.64 | <0.001 |
| No insurance/unknown | 481 (13%) | 256 (4%) | 0.52 | 0.44,0.62 | <0.001 |
| Private | 1,077 (30%) | 1,101 (16%) | REF | REF | REF |
| **Race** |  |  |  |  |  |
| Hispanic | 1,356 (38%) | 1,418 (20%) | 0.39 | 0.35,0.43 | <0.001 |
| Black | 703 (20%) | 2,055 (30%) | 1.08 | 0.96,1.21 | 0.20 |
| Other | 528 (15%) | 724 (10%) | 0.50 | 0.44,0.58 | <0.001 |
| Non-Hispanic White | 1,011 (28%) | 2,746 (40%) | REF | REF | REF |
| **Medication history†** |  |  |  |  |  |
| ACE inhibitor | 0 (0%) | 1,831 (26%) | N/A | N/A | N/A |
| ARB | 0 (0%) | 1,476 (21%) | N/A | N/A | N/A |
| Beta blocker | 0 (0%) | 2,982 (43%) | N/A | N/A | N/A |
| Calcium channel blocker | 0 (0%) | 2,509 (36%) | N/A | N/A | N/A |
| Diuretic | 0 (0%) | 1,975 (28%) | N/A | N/A | N/A |
| Other antihypertensive | 0 (0%) | 681 (10%) | N/A | N/A | N/A |
| Antiplatelet agent | 230 (6%) | 2,880 (41%) | 10.38 | 9.00,12.02 | <0.001 |
| Anticoagulant | 161 (4%) | 1,096 (16%) | 4.00 | 3.37,4.78 | <0.001 |
| **Comorbidities†** |  |  |  |  |  |
| Cardiovascular disease | 389 (11%) | 2,971 (43%) | 6.17 | 5.49,6.94 | <0.001 |
| Hypertension | 1,030 (29%) | 5,946 (86%) | 14.86 | 13.46,16.43 | <0.001 |
| Diabetes | 804 (22%) | 3,423 (49%) | 3.38 | 3.08,3.71 | <0.001 |
| Cancer | 342 (10%) | 933 (13%) | 1.48 | 1.29,1.69 | <0.001 |
| Chronic kidney disease | 192 (5%) | 1,404 (20%) | 4.50 | 3.84,5.29 | <0.001 |
| Dyslipidemia | 529 (15%) | 3,674 (53%) | 6.52 | 5.87,7.25 | <0.001 |
| Obesity (BMI >30 kg/m2) | 1,364 (38%) | 2,968 (43%) | 1.22 | 1.13,1.33 | <0.001 |
| Smoking or vaping | 242 (7%) | 479 (7%) | 1.03 | 0.87,1.21 | 0.78 |
| Immune disorder | 144 (4%) | 360 (5%) | 1.31 | 1.07,1.61 | 0.007 |
| Pulmonary disease | 466 (13%) | 1,461 (21%) | 1.79 | 1.60,2.01 | <0.001 |
| Other comorbidities | 21 (1%) | 5 (0%) | 0.12 | 0.04,0.33 | <0.001 |
| **Outcomes** |  |  |  |  |  |
| Intensive care unit | 950 (26%) | 2,204 (32%) | 1.30 | 1.18,1.42 | <0.001 |
| Mechanical ventilation | 657 (18%) | 1,385 (20%) | 1.12 | 1.01,1.24 | 0.04 |
| Death or hospice | 623 (17%) | 1,589 (23%) | 1.42 | 1.28,1.57 | <0.001 |
| Severe disease\*\* | 1,264 (35%) | 2,897 (42%) | 1.32 | 1.22,1.44 | <0.001 |

\*Change in odds per 10-year increment in age

†OR for use of statin and/or anti-hypertensive use in those with versus without the indicated characteristic or medication

\*\*Severe disease includes need for intensive care unit or mechanical ventilation, or death/discharge to hospice. ACE = angiotensin-converting enzyme; anti-HTN = anti-hypertensive medication; ARB = angiotensinogen II receptor blocker, BMI = body mass index; CI = confidence interval; OR = odds ratio.

## Propensity-score matched analysis in subjects without a history of CVD/Hypertension

Because the use of statins and anti-HTN is strongly linked to the underlying high-risk conditions for which they are prescribed, we used propensity score matching techniques to

**Table 2. Outcomes by statin and anti-hypertensive medication use.**

|  | Death or D/C to Hospice | Mechanical Ventilation | ICU | Death + ICU + Mechanical Ventilation |
|---|---|---|---|---|
| **Exposure:** |  |  |  |  |
| **Statin alone (n = 763)** | 136 (17.8%) | 137 (18.0%) | 228 (29.9%) | 290 (38.0%) |
| **Anti-HTN alone (n = 2494)** | 562 (22.5%) | 496 (19.9%) | 810 (32.5%) | 1057 (42.4%) |
| **Both (n = 3686)** | 891 (24.2%) | 752 (20.4%) | 1166 (31.6%) | 1550 (42.1%) |
| **Neither (n = 3598)** | 623 (17.3%) | 657 (18.3%) | 950 (26.4%) | 1264 (35.1%) |
| **Overall (n = 10541)** | 2212 (21.0%) | 2042 (19.4%) | 3154 (29.9%) | 4161 (39.5%) |

Anti-HTN = anti-hypertensive medication, D/C = discharge, ICU = intensive care unit

Data are n (% of row)

investigate the use of these medications, separately for patients with and without a history of CVD and/or hypertension. Among the 3,107 patients with no history of CVD nor of hypertension, 595 (20%) were taking statin and/or anti-HTN medications ("exposed" subjects). We used the estimated propensity score predicting medication use to match each exposed subject with up to two unexposed subjects who were similar in hospital site, admission month, history of comorbid conditions, and demographic characteristics. We successfully matched 395 exposed subjects with 615 unexposed subjects. We were unable to find an acceptable match for 200 exposed subjects, all of whom had high propensity scores for taking medication; 1807 patients in the unexposed group were not needed (S1A Fig). Among these "healthy" matched patients using medication with neither CVD nor hypertension, use of statin and/or anti-HTN was associated with a 16% lower odds of death (adjusted odds ratio [aOR] 0.84 95% CI 0.58–1.22), but this association was not statistically significant (Fig 2). Medication use was also associated with an 8% lower odds of severe outcome (aOR 0.92, 95% CI 0.70–1.20), which again was not significant.

## Propensity-score matched analysis in subjects with a history of CVD/Hypertension

Among 7,524 patients with a history of CVD and/or hypertension, 6,348 (83%) were taking statins and/or anti-HTN and 1,176 were not. Using a similar propensity score approach as above, 1,124 unexposed patients were each matched with 2,015 exposed patients. A match was available for all but 52 unexposed subjects with low propensity for taking medication, and 4,333 exposed subjects were not needed (S1B Fig). Among subjects with CVD and/or hypertension who were not taking medication, use of statin and/or anti-HTN was associated with about a 30% lower odds of death (aOR 0.68, 95% CI 0.58–0.81), and about a 20% lower odds of a severe outcome (aOR 0.80, 95% CI 0.69–0.93; Fig 2).

## Statins, anti-htn and multivariable association with death and severe disease in all subjects

A multivariable mixed-effects logistic regression model was used to assess the association between medication use and all-cause death, adjusting for patient characteristics, presence of comorbid conditions, potential time trends in disease severity, and potential differences between treating hospitals (modeled as a random effect) in the study population without stratification. In these adjusted models, use of statins either alone or in combination with anti-HTN was associated with a substantial reduction in the chance death (Fig 3A). Compared to those taking neither class of medication, patients taking a statin alone had a 46% lower odds of death

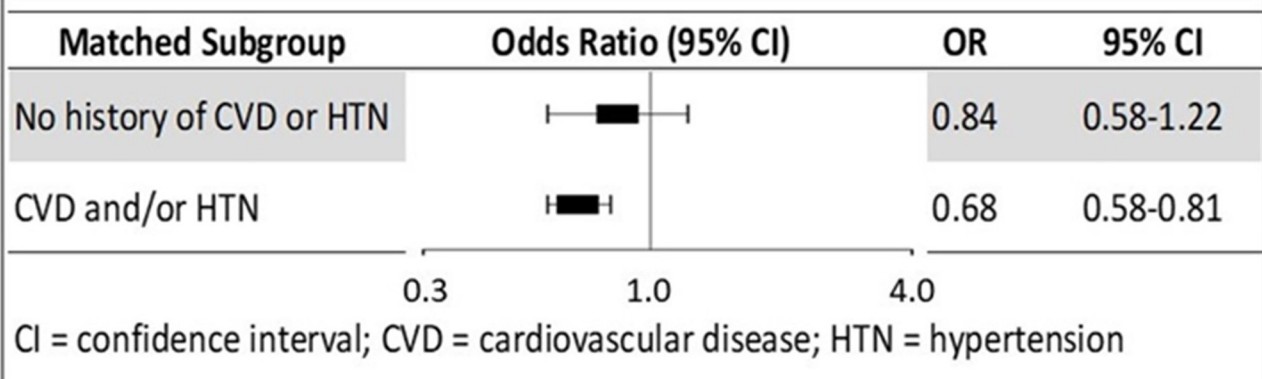

**Fig 2. Odds ratios comparing outcomes for users versus non-users of medications.** Forest plot showing odds ratio comparing outcomes for those using medication (statin and/or anti-hypertensive) versus not using medication in propensity-score matched samples. for (A) death or discharge to hospice, and (B) severe COVID outcome. Analyses done separately for healthy subjects and for those with a history of cardiovascular disease and/or hypertension.

(aOR 0.54, 95% CI 0.43–0.69) while those on both statin and anti-HTN had a 40% lower odds (aOR 0.60, 95% CI 0.50–0.71). Patients with any statin use (alone or in combination with anti-HTN) had a 41% lower odds of death (aOR 0.59, 95% CI 0.50–0.69). Use of anti-HTN alone was associated with a smaller albeit still substantial 27% lower odds (aOR 0.73, 95% 0.62–0.87). There was no significant difference in effect between use of statin alone compared to statin plus anti-HTN (p-value for difference, 0.40). Use of anti-HTN alone was associated with a significantly smaller effect than in combination with statin (p-value for difference, 0.006).

Use of statins and anti-HTN were also associated with a reduction in severe COVID-19 (Fig 3B). Compared to those on neither class of medication, patients taking a statin experienced an approximately 25% lower adjusted odds of developing severe disease (both statin and anti-HTN: aOR 0.73, 95% CI 0.63–0.84; statin alone: aOR 0.74, 95% CI 0.61–0.89; any statin

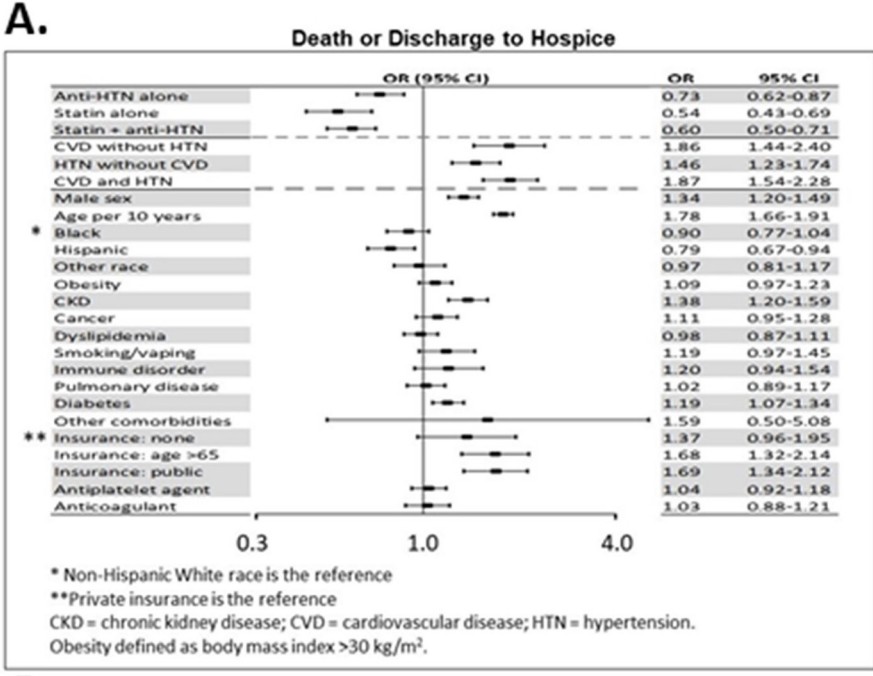

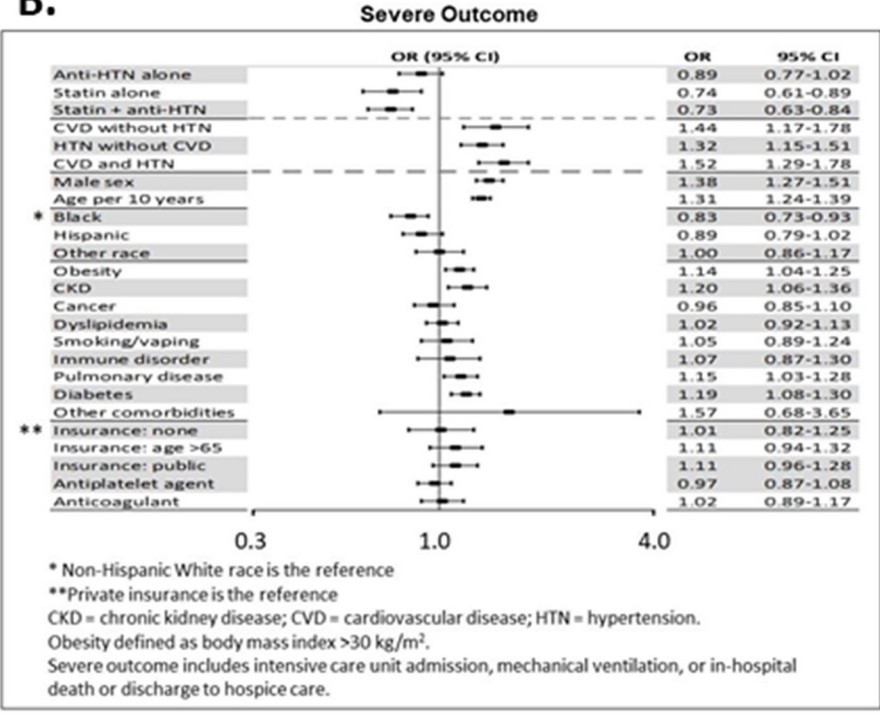

**Fig 3. Multivariable predictors of outcomes.** Predictors of (A) death or discharge to hospice, and (B) severe outcome, in a multivariable logistic regression model.

use, aOR 0.73, 95% CI 0.64–0.84), while patients taking anti-HTN alone had an 11% lower odds of severe disease (aOR 0.89, 95% CI 0.77–1.02).

As a sensitivity analysis, competing-risk analysis to evaluate time to severe outcomes was performed. Compared to taking neither statin nor anti-HTN, patients taking both classes of

medication had a lower rate of development of severe disease (cause-specific adjusted hazard ratio for severe disease 0.84, 95% CI 0.75–0.94). Further details of the competing risk analysis are shown in the S1 Appendix.

## Other multivariable predictors of death and severe disease

Comorbid conditions were generally associated with increased risk of death in adjusted analyses. In particular, compared to patients without hypertension or CVD, patients with CVD had more than an 86% increase in odds of death, with or without the presence of hypertension (CVD alone, aOR 1.86, 95% CI 1.44–2.40; both CVD and hypertension aOR 1.87, 95% CI 1.54–2.28). Those with hypertension alone had an aOR of 1.46 (95% CI 1.23–1.74) compared to patients with neither hypertension nor CVD. Both comorbidities were also associated with risk of severe COVID-19.

Considering other potential confounders, the random effect for hospital site was significant (p-value <0.01), indicating evidence for differences in death rates between sites. Demographic covariates associated with increased risk after adjusting for other covariates were male sex (aOR 1.34, 95% CI 1.20–1.49) and older age (aOR per 10 years 1.78, 95% CI 1.66–1.91). For comorbidities, CKD (aOR 1.38, 95% CI 1.20–1.59) and diabetes (aOR 1.19, 95% CI 1.07–1.34) were also associated with higher risk. The adjustment for time was statistically significant.

## Discussion

In this analysis of over 10,000 subjects hospitalized for COVID-19 across the U.S., use of statins prior to admission was associated with a greater than 40% reduction in mortality and a greater than 25% reduction in risk of developing a severe outcome, after controlling for other medication use, comorbid conditions, hospital site and month of admission, and patient demographic characteristics. The magnitude of this risk reduction was larger than seen for use of anti-hypertensive medications alone.

Because CVD and hypertension are both prominent risk factors for developing severe COVID-19 and are also conditions commonly treated with statins and anti-hypertensive medications, there is a complex interplay between the effects of these conditions and medications. Most patients had CVD, hypertension, or both conditions, and the large majority of these patients were taking at least one relevant medication; for patients with neither condition, the proportion was less than 20%. Use of both medication classes was common. We attempted to disentangle these interactions by using propensity-score matched analyses stratified by comorbidity status. We found that the risk reduction from statins and/or anti-HTN medication use was pronounced among individuals with underlying CVD and/or hypertension, and was less apparent among healthier users of these medications. This is consistent with the hypothesis that the major benefit of these medications accrues from treating and/or stabilizing underlying disease. Although it is well known that statins improve long-term outcomes among patients with or at elevated risk for CVD, the association with a large short-term benefit which accrues in the setting of hospitalization for COVID-19 is a new and intriguing finding.

There are several plausible mechanisms whereby statins could directly mitigate outcomes in COVID-19, beyond treating underlying disease conditions [11]. Organ dysfunction in COVID-19, including acute respiratory distress syndrome, myocardial injury, renal injury, and a hypercoagulable state has been attributed to severe inflammation and the subsequent cytokine storm and pervasive endothelial dysfunction [12–14]. Death as a result of cardiovascular injury in COVID-19 has been thought to be a result of the inflammatory cascade and subsequent plaque rupture following destabilization [15–17]. Statin therapy has been shown to decrease inflammatory markers and provide plaque stabilization effects within coronary

vasculature, irrespective of elevated cholesterol levels [18, 19]. Some observational studies have also found statin therapy to be associated with improved survival in patients with community acquired pneumonia [20], and with reduced hospitalization and mortality in influenza, presumably through inflammatory pathway modulation [7, 21–23]. Statins may also have a direct inhibitory effect on SARS-CoV2 [6]. In addition, a recent study showed that SARS-CoV-2 induces cholesterol 25-hydroxylase (25HC) both *in vitro* and in COVID-19 infected patients, via interferon signaling [24]. 25HC activation leads to depletion of accessible cholesterol on cell membranes and results in broad anti-coronavirus activity by blocking viral-cell fusion and preventing viral infection of lung epithelial cells [24]. Statins may similarly deplete cholesterol from cell membranes resulting in coronavirus suppression. However, given that only modest effects on disease severity were seen among patients without underlying CVD or hypertension, any such direct effects may be of less importance in previously healthy patients.

Our results are consistent with most prior studies, though these most of these have been small or regional. A single-center U.S. observational study of 170 subjects found a 50% reduction in severe COVID-19 among patients with statin use prior to hospitalization, consistent with our findings [1]. A larger, regional study from Hubei Province, China described an approximately 40% reduction in mortality with statin use, however it looked at inpatient use of statins and may suffer from bias by treatment indication as well as cofounding from lack of adjustment for prehospital use of statins and socioeconomic factors; further, fewer than 10% of subjects in that study were on statins [2]. Similarly, a study based on a U.S. hospital claims database that covered 21,676 hospitalizations with any COVID-19 diagnosis across 276 hospitals, found that in-hospital statin use was associated with a 46% reduction in in-hospital mortality [25]. A few studies have failed to find an association between statin use and COVID-19 severity, however these were either very small studies, or based upon administrative data which can be less granular. For instance, a Danish study of 4,842 subjects did not find an association between statin use and outcomes, however this study included any patient with an ICD-10 code signifying a COVID-19 diagnosis, regardless of hospitalization status, and had a prevalence of statin use of only 17% [5]. In contrast, in the present inpatient, racially diverse, large nationwide cohort, 42% of subjects were on statins, and over 4,100 severe outcomes were analyzed including 2,212 deaths. Another outlier was the Coronavirus–SARS-CoV-2 and Diabetes Outcomes (CORONADO) study of 2,449 patients with COVID-19 and type 2 diabetes mellitus (including 1,192 statin users and about 513 deaths) who were admitted to one of 68 French hospitals during a very early month of the pandemic. This study found that inpatient statin use was associated with a higher risk of mortality in propensity-matched analyses. However, this finding of increased risk has not been replicated in other observational studies of inpatients with diabetes [26, 27].

Our study has several strengths, most notably the large study population with large numbers of both medication exposures and documented outcomes which was critical in enabling us to address the significant confounding relationships between statin and anti-HTN use, and underlying CVD and hypertension. To our knowledge, this is the first analysis to investigate the effect of statins on COVID-19 outcomes in a "healthy" population without underlying CVD or hypertension. Although the effect of statins in this healthier population was not statistically significant, the point estimates of the odds ratios suggest that statins could still be protective. Other major strengths include the demographic diversity of the patient population and geographic diversity of hospital sites, and the detailed, individually extracted patient-level information on comorbidities, insurance status, and other confounders. The data abstraction algorithms and quality control and data checks in the AHA COVID-19 Cardiovascular Disease Registry are well documented.

Our study also has several limitations of note. As an observational study, this analysis is unable to prove causality. We attempted to account for confounders with both multivariable models and with propensity-score matched analyses, but the possibility of residual confounding remains. Misclassification is also a possibility, though we attempted to limit this bias by excluding patients who were transferred from outside facilities, in whom outpatient medication regimens are often more difficult to verify. We were unable to evaluate the in-hospital use of statin and anti-HTN medications, nor the association with specific statins or doses, as these data were not collected. In addition, we did not have data on the specific indication for the medications studied. The present study, which includes data from January through September 2020, does not include the latest, largest surge in COVID-19 hospitalizations, from which additional patterns and associations may become apparent and better defined. Also, study outcomes were recorded by study personnel at local sites without central adjudication; however the outcomes we chose, including in-hospital death, are endpoints not generally subject to major recording bias.

Our findings have important clinical implications. Early in the pandemic there was speculation that certain medications, including statins, ACE inhibitors and ARBs, could confer an increased susceptibility to COVID-19 positivity and/or severity. Our study reinforces the AHA and others' recommendations that not only is it safe to remain on these medications, but they may substantially reduce risk of severe COVID-19 and especially death from COVID-19 [28], particularly statins, and particularly among those with associated underlying conditions. Whether these medications are also protective among individuals without underlying indications for taking them such as CVD or hypertension is less clear and merits further study. Several randomized trials are currently underway evaluating the use of these medications for treatment of COVID-19 [29–31].

## Conclusion

Use of statins prior to hospitalization for COVID-19 is associated with a substantially reduced risk of death and severe COVID-19, especially among those with CVD or hypertension. The benefit in patients without these underlying conditions appears to be less pronounced. Randomized, controlled trials are underway to further elucidate the role that statins may have in the treatment of COVID-19.

## Supporting information

**S1 Appendix. Competing risk sensitivity analysis for time to severe outcomes.**
(PDF)

**S1 Table. Characteristics of patients who survived compared with those with in-hospital death/discharge to hospice A.**
(PDF)

**S2 Table. Characteristics of patients with severe versus non-severe COVID-19.**
(PDF)

**S1 Fig. Plot of propensity scores by matching status.** Plot of propensity scores by matching status for (A) "healthy" patients, with neither a history of cardiovascular disease nor hypertension, and (B) patients with a history of cardiovascular disease and/or hypertension. Exposed versus unexposed refers to history of statin and/or anti-hypertensive medication use prior to hospitalization. The y axis shows the estimated probability of medication use (i.e. the propensity score), from the propensity score logistic regression model. Tables show the odds ratio (OR) for risk of death (in-hospital death or discharge to hospice) or severe disease (need for

intensive care unit or mechanical ventilation, or death), comparing exposed to matched unexposed patients.
(TIF)

## Author Contributions

**Conceptualization:** Lori B. Daniels, Quan M. Bui, Mariem A. Sawan, Howard Eisen, Christopher A. Longhurst, Karen Messer.

**Data curation:** Lori B. Daniels.

**Formal analysis:** Junting Ren, Jing Zhang, Xinlian Zhang, Karen Messer.

**Investigation:** Lori B. Daniels, Kris Kumar.

**Methodology:** Lori B. Daniels.

**Project administration:** Lori B. Daniels.

**Resources:** Lori B. Daniels, Christopher A. Longhurst.

**Software:** Christopher A. Longhurst.

**Supervision:** Lori B. Daniels.

**Writing – original draft:** Lori B. Daniels, Junting Ren, Kris Kumar, Quan M. Bui, Karen Messer.

**Writing – review & editing:** Lori B. Daniels, Junting Ren, Kris Kumar, Quan M. Bui, Jing Zhang, Xinlian Zhang, Mariem A. Sawan, Howard Eisen, Christopher A. Longhurst, Karen Messer.

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
