## [Decision Letter · Decision Letter 0]

14 Jun 2021

PONE-D-21-16257

Relation of prior statin and anti-hypertensive use to severity of disease among patients hospitalized with COVID-19: Findings from the American Heart Association’s COVID-19 Cardiovascular Disease Registry

PLOS ONE

Dear Dr. Daniels,

Thank you for submitting your manuscript to PLOS ONE. After careful consideration, we feel that it has merit but does not fully meet PLOS ONE’s publication criteria as it currently stands. Therefore, we invite you to submit a revised version of the manuscript that addresses the points raised during the review process.

We look forward to receiving your revised manuscript.

Kind regards,

Aleksandar R. Zivkovic

Academic Editor

PLOS ONE

3. Please upload a copy of Supporting Information Figure 1 which you refer to in your text on page 24.

**Comments to the Author**

Reviewer #1: The authors provide a timely and relevant study. The most important findings from the study are that hospitalized COVID-19 patients with cardiovascular disease or hypertension were more likely to die during their hospitalization, but statin therapy lowered their odds of death. The strengths of the study are the large number of patients and the well thought out and thorough design of the study. The manuscript is well written and the conclusions are presented in a clear and concise fashion.

One point that needs to be addressed is how the authors discerned hypertensive medications from the use of these medications for other reasons. For example a patient may have been on a beta-blocker or ACE inhibitor for heart failure not for hypertension. The authors should have cross referenced ICD codes or a history of other conditions that could be the reason why the patients were on these medications. If patients on these medications were excluded if they had a diagnosis of heart failure, angina, or arrhythmia would the use of antihypertensive medications still have shown a clinical benefit? In addition in the methods section how were the comorbidities defined? For example was CKD defined by estimated GFR or by history and the same for hyperlipidemia?

Reviewer #2: Thank you for this carefully performed analysis on a highly interesting and constantly progressing area. This large cohort adds to the literature and I don't have relevant comments to improve the analysis as is. However, a few points/questions for your consideration:

1. There are constantly new reports, recently a review by EuGMS Special Interest group on CVD listed (in its appendix) observational studies on statins and mortality. This reference might be used (Alves et al. Eur Geriatr Med 2021 2021 May 25:1-15. doi: 10.1007/s41999-021-00504-5) to emphasize totality of evidence in observational studies in the absence of RCTs.

2. Of larger studies on the topic the CORONADO study -- or rather its critique (Strandberg TE, Kivimäki M. Diabetes Metab. 2021 May;47(3):101250. doi: 10.1016/j.diabet.2021.101250) -- could be mentioned, because CORONADO with its outlier (and doubtful) results is nevertheless frequently quoted.

3. It might be polite to give credit to Dr David Fedson, because he has been a pioneer in emphasizing the host condition and potential benefits of the use of statins and RAAS drugs in serious infections, already before the present pandemic (David S Fedson. Influenza, evolution, and the next pandemic. Evol Med Public Health 2018 Oct 3;2018(1):260-269. doi: 10.1093/emph/eoy027. eCollection 2018).

4. In all observational studies like this the authors state that "RCTs are needed". But what does that actually mean? If statins work acutely in an infection, an RCT makes sense. But if the mechanism is that prior/ongoing statin treatment protects and gives prior resilience to the covid-19 patient, which I believe is most likely, what would the outline of the RCT be to prove that mechanism. Trial should randomise patients 1/2-1 year previously and prospectively follow-up them. Would such trial likely to be realized? Not likely.

6. PLOS authors have the option to publish the peer review history of their article (what does this mean?). If published, this will include your full peer review and any attached files.

Reviewer #1: **Yes: **Dean Karalis

Reviewer #2: **Yes: **Timo E Strandberg

---

## [Author Response · Author response to Decision Letter 0]

27 Jun 2021

(Please also see attached file, entitled 'Response to Reviewers', and cover letter.)

Response to Reviewers – Rebuttal Letter

We thank each of the reviewers for their careful reading of our manuscript and their insightful and constructive comments. We have considered each of the suggestions and comments in some detail, and we respond point-by-point below. We have also made each of the technical edits as requested. We hope that the editors and reviewers are satisfied that we have successfully addressed the comments, and that you now find this manuscript suitable for publication. Thank you again for the careful and helpful reviews.

Reviewer #1:

The authors provide a timely and relevant study. The most important findings from the study are that hospitalized COVID-19 patients with cardiovascular disease or hypertension were more likely to die during their hospitalization, but statin therapy lowered their odds of death. The strengths of the study are the large number of patients and the well thought out and thorough design of the study. The manuscript is well written and the conclusions are presented in a clear and concise fashion.

1. One point that needs to be addressed is how the authors discerned hypertensive medications from the use of these medications for other reasons. For example a patient may have been on a beta-blocker or ACE inhibitor for heart failure not for hypertension. The authors should have cross referenced ICD codes or a history of other conditions that could be the reason why the patients were on these medications. If patients on these medications were excluded if they had a diagnosis of heart failure, angina, or arrhythmia would the use of antihypertensive medications still have shown a clinical benefit?

The reviewer is correct that discerning the reason that patients are on a particular medication is not possible in this retrospective observational study. Unfortunately, our dataset is strictly limited by the data that were collected with the data collection forms used by the American Heart Association’s Get With The Guidelines CVD COVID-19 Registry. This did not include ICD codes; but did include reported past medical history of various cardiovascular conditions, including hypertension, coronary artery disease, and heart failure. We were able to use this reported history of other conditions to conduct our propensity score analyses, in which we evaluated patients without a reported history of hypertension or cardiovascular disease. However, we were unable to evaluate these conditions individually (i.e. limit the dataset to those with hypertension while excluding those with other cardiovascular diseases) due to concern for misclassification (i.e. being labelled as ‘hypertensive’ based upon taking anti-hypertensives, even if the medication was prescribed for another indication such as heart failure) since we did not have data on the specific indication for the medications studied. This is a limitation of the current study, and we have now added this to our Discussion section, as follows (Discussion, p19): 

“In addition, we did not have data on the specific indication for the medications studied.”

2. In addition in the methods section how were the comorbidities defined? For example was CKD defined by estimated GFR or by history and the same for hyperlipidemia?

The case report form had pre-specified definitions for each comorbidity. CKD was defined as a history of physician diagnosed renal insufficiency or chronic failure or if the serum creatinine was greater than 2.0mg/dl. Dyslipidemia was defined as a history of high cholesterol, hyperlipidemia or hypercholesterolemia based on physician diagnosis, treatment with a lipid lowering agent, total cholesterol greater than 200, LDL greater than 100, HDL less than 40, or elevated triglycerides greater than 200. Diabetes was defined as a history of confirmed physician diagnosed diabetes mellitus (type I or II), or treatment for diabetes including the use of diet, oral hypoglycemic agents or insulin. In addition, hypertension (HTN) was defined as a history of physician-diagnosed high blood pressure, regardless of treatment.

We have now added this information to the Methods section.

3. The authors should specify the rate of death and admission to ICU.

We have now included the following information (Results, p8):

“The primary study population included 10,541 subjects hospitalized with COVID-19 disease, including 2,212 (21%) who died and 4,161 (39%) who experienced a severe outcome. ICU admission incidence overall was 30%, and 19% required mechanical ventilation.”

Reviewer #2:

Thank you for this carefully performed analysis on a highly interesting and constantly progressing area. This large cohort adds to the literature and I don't have relevant comments to improve the analysis as is. However, a few points/questions for your consideration:

1. There are constantly new reports, recently a review by EuGMS Special Interest group on CVD listed (in its appendix) observational studies on statins and mortality. This reference might be used (Alves etal. Eur Geriatr Med 2021 2021 May 25:1-15. doi: 10.1007/s41999-021-00504-5) to emphasize totality of evidence in observational studies in the absence of RCTs.

We thank the Reviewer for pointing out this excellent and thorough review, which is indeed a great reference for this rapidly changing field. We have now updated our references and Discussion accordingly.

2. Of larger studies on the topic the CORONADO study -- or rather its critique (Strandberg TE, KivimäkiM. Diabetes Metab. 2021 May;47(3):101250. doi: 10.1016/j.diabet.2021.101250) -- could be mentioned, because CORONADO with its outlier (and doubtful) results is nevertheless frequently quoted.

Again, we thank the Reviewer for pointing this out, and we have now including this in our Discussion, as follows:

“Another outlier was the Coronavirus–SARS-CoV-2 and Diabetes Outcomes (CORONADO) study of 2,449 patients with COVID-19 and type 2 diabetes mellitus (including 1,192 statin users and about 513 deaths) who were admitted to one of 68 French hospitals during a very early month of the pandemic. This study found that inpatient statin use was associated with a higher risk of mortality in propensity-matched analyses. However, this finding of increased risk has not been replicated in other observational studies of inpatients with diabetes.[24, 25]”

3. It might be polite to give credit to Dr David Fedson, because he has been a pioneer in emphasizing the host condition and potential benefits of the use of statins and RAAS drugs in serious infections, already before the present pandemic (David S Fedson. Influenza, evolution, and the next pandemic. Evol Med Public Health 2018 Oct 3;2018(1):260-269. doi: 10.1093/emph/eoy027. eCollection 2018).

We thank the reviewer for pointing out this oversight, as we agree that Dr. Fedson had some of the early pioneering work in this field, and indeed we have read many of his manuscripts. We have now remedied this and included his work in our references. 

4. In all observational studies like this the authors state that "RCTs are needed". But what does that actually mean? If statins work acutely in an infection, an RCT makes sense. But if the mechanism is that prior/ongoing statin treatment protects and gives prior resilience to the covid-19 patient, which I believe is most likely, what would the outline of the RCT be to prove that mechanism. Trial should randomise patients 1/2-1 year previously and prospectively follow-up them. Would such trial likely to be realized? Not likely.

We agree that it would be nearly impossible to conduct such a randomized trial to study the use of statins (or ACEi/ARBs) months prior to infection, especially now with vaccinations highly prevalent; and this is likely to be an important mechanism by which resilience to COVID-19 is realized. However, treatment in the acute phase of the illness has also been posited to be potentially useful, and this is indeed being studied in a wide variety of randomized, controlled trials. (Very recently, a small study published online in Lancet E-Clinical Medicine showed that high-dose telmisartan, started early in the course of acute COVID-19, may have beneficial effects [https://www.sciencedirect.com/science/article/pii/S258953702100242X]; and trials of statins have been undertaken as well.) Having said this, we also agree that it is somewhat trite to always state that randomized trials are needed; alas, this is often true, because observational studies are subject to so much potential confounding and unfortunately remain primarily hypothesis-generating. Nonetheless, we were intentionally careful with our wording – we believe that the present study is certainly sufficient to reassure patients that staying on their medications despite the pandemic is at the very least safe – and it is almost certainly remains beneficial for those with underlying cardiovascular conditions.

---

## [Editor Report · Decision Letter 1]

1 Jul 2021

Relation of prior statin and anti-hypertensive use to severity of disease among patients hospitalized with COVID-19: Findings from the American Heart Association’s COVID-19 Cardiovascular Disease Registry

PONE-D-21-16257R1

Dear Dr. Daniels,

We’re pleased to inform you that your manuscript has been judged scientifically suitable for publication and will be formally accepted for publication once it meets all outstanding technical requirements.

Kind regards,

Aleksandar R. Zivkovic

Academic Editor

PLOS ONE

---

## [Editor Report · Acceptance letter]

5 Jul 2021

PONE-D-21-16257R1 

Relation of prior statin and anti-hypertensive use to severity of disease among patients hospitalized with COVID-19: Findings from the American Heart Association’s COVID-19 Cardiovascular Disease Registry 

Dear Dr. Daniels:

I'm pleased to inform you that your manuscript has been deemed suitable for publication in PLOS ONE. Congratulations! Your manuscript is now with our production department. 

Kind regards, 

on behalf of

Dr. Aleksandar R. Zivkovic 

Academic Editor

PLOS ONE